# Microbiological Analysis and Metagenomic Profiling of the Bacterial Community of an Anthropogenic Soil Modified from Typic Haploxererts

Pietro Barbaccia [1], Carmelo Dazzi [1], Elena Franciosi [2], Rosalia Di Gerlando [1], Luca Settanni [1,*] and Giuseppe Lo Papa [1]

1   Dipartimento Scienze Agrarie, Alimentari e Forestali, Università degli Studi di Palermo, Viale delle Scienze 4, 90128 Palermo, Italy; pietro.barbaccia@unipa.it (P.B.); carmelo.dazzi@unipa.it (C.D.); rosalia.digerlando@unipa.it (R.D.G.); giuseppe.lopapa@unipa.it (G.L.P.)
2   Research and Innovation Centre, Fondazione Edmund Mach (FEM), 38098 San Michele all'Adige, Italy; elena.franciosi@fmach.it
*   Correspondence: luca.settanni@unipa.it

**Abstract:** This work aimed to characterize the microbial communities of an anthropogenic soil originating from application of pedotechniques to Vertisols in a Mediterranean environment. Bare soil profiles were sampled at three depths (0–10 cm, 10–30 cm, and 30–50 cm) and compared with the original soil not transformed at the same depths. The anthropogenic soils were characterized by a higher $CaCO_3$ concentration (360–640 g/kg) than control soil (190–200 g/kg), while an opposite trend was registered for clay, where control soil showed a higher concentration (465 g/kg on average) than anthropogenic soil (355 g/kg on average). Organic carbon content was much higher in the untransformed soil. All samples were microbiologically investigated using a combined culture-dependent and -independent approach. Each pedon displayed a generally decreasing level with soil depth for the several microbial groups investigated; in particular, filamentous fungi were below the detection limit at 30–50 cm. To isolate bacteria actively involved in soil particle aggregation, colonies with mucoid appearance were differentiated at the strain level and genetically identified: the major groups were represented by *Bacillus* and *Pseudomonas*. MiSeq Illumina analysis identified Actinobacteria and Firmicutes as the main groups. A high microbial variability was found in all the three anthropogenic pedons and the microorganisms constitute a mature community.

**Keywords:** anthropogenic soil; applied soil ecology; extracellular polymeric substances; MiSeq Illumina; viable bacteria

## 1. Introduction

In recent decades, ex novo soil formation by human action has become increasingly important to crop-linked activities [1]. Human activity is recognized as one of the most important factors involved in soil generation in the 21st century [2,3]. Generally, the term "Anthrosol" refers to soil formed or strongly modified by human activity such as deep tillage, intensive fertilization, addition of organic waste, irrigation with water rich in sediments or creation of rice fields [4].

Studies focusing on the genetic peculiarities, characteristics and properties of soils created in urban areas, mines, forested and agricultural areas evidence that the main issue of the pedogenetic process is the loss of the soil structure [2,3,5–7]. In particular, the agricultural areas of southern Italy are characterized by a continuous loss of structure as a result of deep tillage and displacement of material. This phenomenon, combined with semi-arid conditions and poor vegetation cover, determines runoff and wind erosion during seasonal rainfall, and in the long term, reduces soil fertility [8].

Organic and inorganic input, cover crops and deep tillage have a negative influence on the aggregation status of the soil particles, while microorganisms play an important role

in soil aggregation [9,10]. Aside from improving the structure of soil, the soil microbiome plays a key role in several functional processes such as production of substances for plant growth promotion, degradation of pollutants derived from agrochemical treatment and control of all type of pests [11]. Soil microbial diversity is extremely complex; hundreds of thousands of microbial taxa live together in a gram of soil [12], with a total number of microorganisms exceeding 10 billion [13].

Prokaryotic organisms, both bacteria and archaea, can live and develop within biofilms consisting of extracellular polymeric substances (EPS). These substances are mainly formed by exopolysaccharides structural proteins, enzymes, biopolymeric substances such as lipids and other constituents [14,15]. In natural environments, most microorganisms live in aggregates such as floccules or biofilms, of which EPSs represent the fundamental structural component [16]. In a biofilm, the microbial cells are embedded within a self-produced EPS matrix. This organization allows microorganisms to adhere to each other and/or to a surface. Specifically, "a biofilm is a fixed system that can be adapted internally to environmental conditions by its inhabitants" [17]. This structure represents a real selective ecological advantage [16]. Thanks to biofilms, several bacteria adhere to and communicate with other microorganisms and plants. A given biofilm is characterized by water channels that allow for the passage of nutrients and other agents throughout the bacterial community [18]. The water channels within biofilms are considered as a primitive circulatory system which protects bacteria against the accumulation of toxic metabolites by their removal [19]. Biofilms constitute real barriers that protect microorganisms from predation, anti-microbial substances and heavy metals, and adverse environmental stresses [15,20,21].

The formation of structural aggregates in soil can be greatly improved by the bacterial EPS. These substances positively affect the structure, porosity, fertility and productivity of the soil systems [22,23]. EPSs can act as glues due to their viscous texture and ionic charges that allow for anchoring to soil clays [24], thus improving the aggregation of poorly aggregated soil particles [16].

The current study was undertaken (1) to assess the soil microbial community and (2) to investigate specifically EPS-producing populations of an anthropogenic soil prior to its first cultivation cycle. Specifically, the microbial composition of the sampled soil was approached by culture-dependent and -independent methods. The soil was also characterized for its physical and chemical properties.

## 2. Materials and Methods

### 2.1. Study Area

The study area (Figure 1) is located in the countryside of Palma di Montechiaro—the Giordano district in the province of Agrigento (AG) in southern Sicily—and is characterized by a Mediterranean climate. This hilly agricultural area is also characterized by the presence of several horticultural greenhouse farms, mainly cultivating table grapes, obtained through anthropogenic interventions on soils. The lithology dates back to the Pliocene and Miocene periods and consists of gypsum, marls, limestones and alluvial deposits. The soils are classified as Entisols, Inceptisols, Mollisols and Vertisols.

Soil sampling occurred in March 2019. Climatic data for the reference year were obtained from the Agrigento meteorological station. The average annual temperature was 17.7 °C, with the highest thermometric value (41.5 °C) registered in July and the lowest (−1.0 °C) in January. The average annual precipitation was 497 mm.

### 2.2. Applied Pedotechnics

A new soil was generated by pedotechnics in 2012. The entire area is characterized by Typic Haploxererts. This soil was covered with calcareous marls in October 2012 and levelled using a caterpillar machine during the following summer. In Autumn 2013, soil was ploughed to an approx. depth of 90–100 cm using a one mouldboard, single-furrow plough. Thereafter, it was left uncultivated.

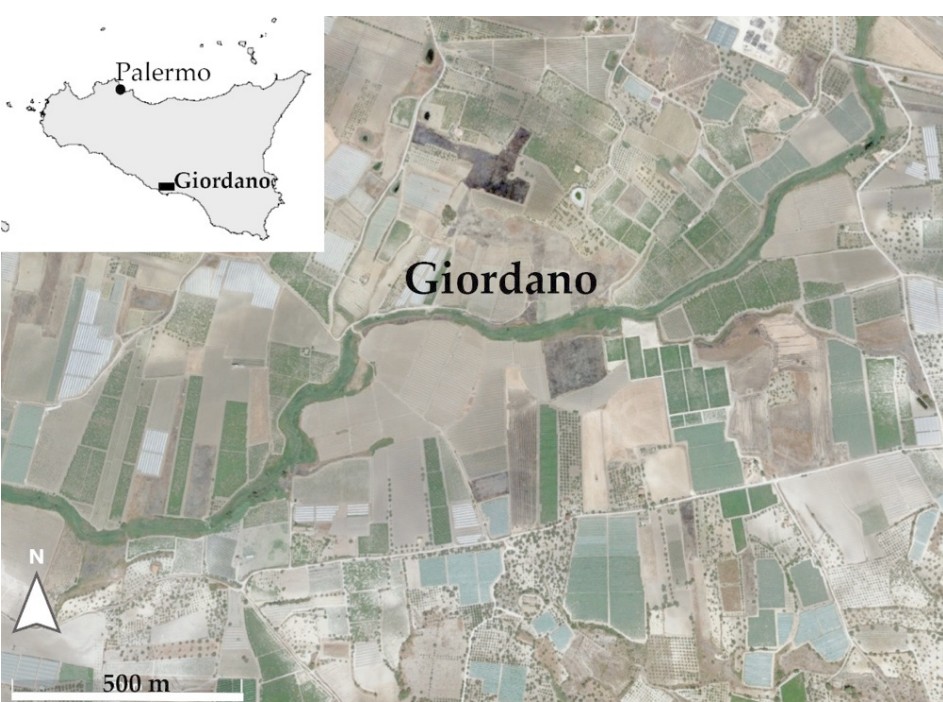

**Figure 1.** Location of the study area.

*2.3. Sampling*

Soil samples for physicochemical and microbiological analyses were collected from the following pedons (Figure 2): Mant 0 (37°10′08.1″ N–13°46′56.8″ E), Mant 1 (37°10′09.3″ N–13°47′01.4″ E), Mant 2 (37°10′09.4″ N–13°47′00.8″ E) and Mant 3 (37°10′09.7″ N–13°47′00.1″ E). Mant 0 refers to the original Vertisol not affected by pedotechnics (control soil), while Mant 1, Mant 2 and Mant 3 concern profound transformations due to the pedotechnics used and represent three technical repeats. Each pedon was sampled in duplicate at 3 different depths: I, 0–10 cm; II, 10–30 cm; and III, 30–50 cm, corresponding to part of Ap and ^A horizons, in Vertisol and anthropogenic soils, respectively.

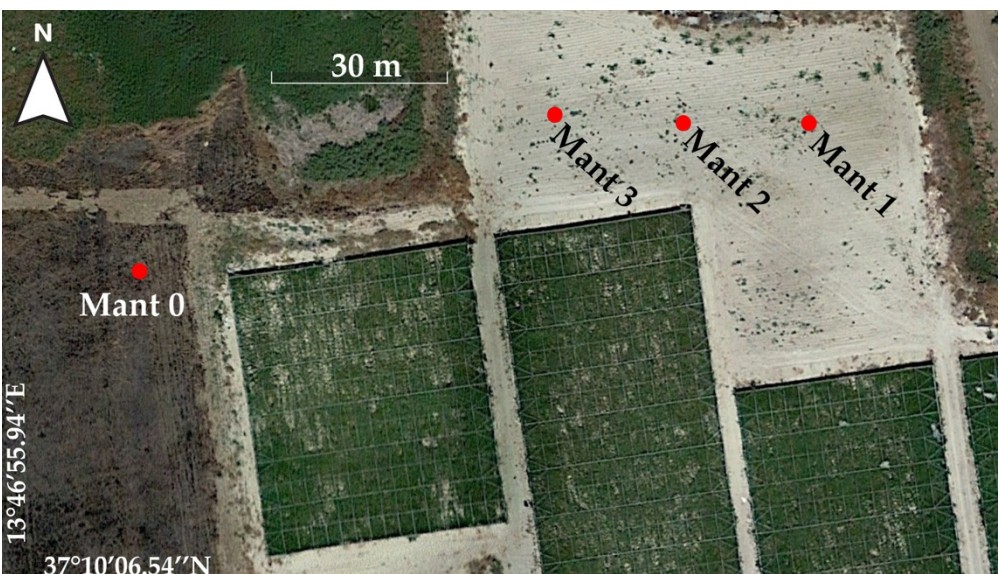

**Figure 2.** Sampling points. Red circles are soil profiles.

## 2.4. Physicochemical Analyses

Soil samples were air-dried and sieved by means of a 2 mm stainless steel sieve (Laboratory Test Sieve, London, UK). Texture was determined with the sedimentation method [25]. pH was measured on soil–water mixtures 1:2.5 (*w/v*) with a pH meter (XS instruments, Carpi, Italy). The electrical conductivity (EC) was determined on each soil–water mixture 1:5 with a portable conductivity-meter Cond 7 (XS instruments, Carpi, Italy). The cation exchange capacity (CEC) was determined by barium chloride and triethanolamine method (MIPAF, 2000). Total carbonates ($CaCO_3$ Tot) were determined by gas volumetric method using a Dietrich–Fruehling calcimeter and HCl, while organic carbon ($C_{org}$) was determined with the wet oxidation method of Walkley and Black [26].

## 2.5. Culture-Dependent Microbiological Analysis

Soil samples for microbiological analysis were collected in aseptic conditions: a sterile area was obtained using a portable Bunsen burner, and each sample (approximately 200 g) was picked up with a sterile stainless-steel spatula and transferred into a sterile BagLightR 400 MultilayerR bag (Interscience, Saint Nom, France). All samples were transported under refrigeration in a portable fridge to the Laboratory of Agricultural Microbiology of University of Palermo.

All samples, according to Thomson et al., were dried for 24 h under a laminar hood to avoid environmental contaminations and sieved with the 2 mm sieve after autoclaving (121 °C for 15 min) [27]. All sieved samples were placed into 9 cm diameter Petri dishes, sealed with Parafilm, and kept at room temperature until plate counting and total DNA extraction.

Plate counts were performed to determine the levels of the main soil microbial populations such as total mesophilic aerobic (TMA) microorganisms, total mesophilic anaerobic (TMAn) microorganisms, filamentous fungi (FF), Actinobacteria (AB), N-fixing (NF) bacteria, and the main EPS and glue-producing bacteria, *Sphingomonas* and *Caulobacter*. Soil samples (20 g) were put into conical flasks and diluted (1:10) with sodium pyrophosphate (0.16% *w/v*) solution (180 mL) by homogenization with an orbital shaker for 10 min at 150 rpm [28]. Further decimal dilutions of soil samples were performed in test tubes containing Ringer's solution (Sigma-Aldrich, Milan, Italy) subjected to homogenization by vortexing.

The microbial suspensions were inoculated in different culture media and incubated in the optimal conditions (temperature and time). TMA were inoculated in bacteria medium, incubated at 30 °C for 48 h, while FF were inoculated in fungi medium at 30 °C for 7 d; both media were prepared as described by Zhang et al. [29]. TMAn were inoculated in soil extract medium (SEM), prepared as described by Deutsche Sammlung von Mikroorganismen und Zellkulturen (https://www.dsmz.de/microorganisms/medium/pdf/DSMZ_Medium12.pdf, accessed on 19 March 2019), incubated at 30 °C for 48 h in hermetically sealed jars containing the AnaeroGen AN25 system (Oxoid, Milan, Italy). *Caulobacter* and *Sphingomonas* were grown in two selective media, *Caulobacter* medium (CM) [30] and NK medium [31] prepared, respectively, and both incubated at 30 °C for 48 h. NF bacteria were inoculated in Blue Green Medium (BG-11) prepared as described by Rippka et al., and the plates were incubated at 30 °C for 48 h [32]. All microbiological analyses were carried out in duplicate and the results expressed as Log colony-forming units (CFU) per g of dry weight (g dw), determined after drying at 105 °C until constant weight.

White/yellowish colonies with mucoid appearance, presumptive *Caulobacter* and *Sphingomonas*, were harvested from CM and NK, grown in Nutrient Broth (NB) (Oxoid) and subjected to purification by consecutive streaking on Nutrient Agar (NA) until reaching morphology homogeneity of colonies. The isolates were first subjected to a phenotypic investigation under an optical microscope to analyze cell morphology. Only rod bacteria were further processed by being genetically investigated, such as *Caulobacter* and rod-shaped *Sphingomonas*.

DNA was extracted after overnight growth of the pure cultures in NB at 30 °C for 48 h. The cells were harvested from 1 mL of broth cultures by centrifugation at 10,000× *g* for 5 min, and the pellets were subjected to repeated washing steps with sterile distilled H$_2$O and finally re-suspended in 1 mL of the same diluent. Cell lysis was performed by Instagene Matrix (Bio-Rad, Hercules, CA, USA) following the protocol provided by the supplier. Crude cell extract from each culture was used as template DNA for the genotypic characterization.

All isolates were differentiated at the strain level to reduce the number of bacteria to be genetically identified. The typing procedure was carried out by random amplification of polymorphic DNA (RAPD)-PCR as described by Gaglio et al. [33]. RAPD profiles were visualized under UV light after electrophoresis of 2% (*w/v*) agarose gels in 1 × TBE buffer (Sigma-Aldrich, Milan, Italy). The isolates that showed diverse RAPD profiles were considered different strains and analyzed by sequencing the ribosomal 16S rRNA gene for species identification, applying the protocol described by Weisburg et al. [34]. PCR products of about 1600 bp were purified using the QIAquick kit (Quiagen S.p.a., Milan, Italy) and sequenced at the center for innovation of quality systems, traceability and certification of agri-food—AGRIVET (University of Palermo), with the same oligonucleotides used for PCR analysis. The identity of each sequence was obtained by comparing the sequences acquired with those available in GenBank/EMBL/DDBJ (http://www.ncbi.nlm.nih.gov, accessed on 19 March 2021) [35] and Ez-Taxon (http://eztaxon-e.ezbiocloud.net/ accessed on 19 March 2021) [36] databases. The latter database compares the sequences obtained with those of the type strains only.

### 2.6. Culture-Independent Analysis

The amount of soil for total genomic DNA extraction from each sample ranged between 0.4 and 0.6 g. DNA extraction was performed by the Gene MATRIX Soil DNA Purification Kit (EURx, Gdansk, Poland) according to the manufacturer's instructions, and quantified using a Nanodrop 8800 Fluorospectrometer (Thermo Fisher Scientific, Waltham, MA, USA).

#### 2.6.1. Amplicon Library Preparation

The pooled libraries and pair-end sequencing quality and quantification were carried out using the sequencing platform of Edmund Mach Foundation (FEM, San Michele a/Adige, Italy). Soil genomic DNAs were amplified with primers specific to the V3–V4 region [37,38] of the 16S rRNA gene of *Escherichia coli* corresponding to the positions 341 to 805. The PCR reaction volume of 25 μL contained 1 μM of each primer. PCR results were obtained through the GeneAmp PCR System 9700 (Thermo Fisher Scientific). Amplicons were checked on 1.5% agarose gel and then purified through the system Agencourt AMPure XP (Beckman Coulter, Brea, CA, USA). A further PCR was necessary to apply dual indices and Nextera XT Index Primer (Illumina, San Diego, CA, USA), which are Illumina sequencing adapters. The resulting libraries were purified as reported above. The libraries were also checked for quality on the Typestation 2200 platform (Agilent Technologies, Santa Clara, CA, USA). Barcoded libraries were pooled in an equimolar ratio and sequenced on the MiSeq Illumina® (PE300) platform (MiSeq Control Software 2.5.0.5 and Real-Time Analysis software 1.18.54.0).

#### 2.6.2. Illumina Data Analysis and Sequences Identification by QIIME2

FASTQ files containing raw paired-end sequences were demultiplexed by idemp (https://github.com/yhwu/idemp/blob/master/idemp.cpp, accessed on 17 December 2021) and imported into Quantitative Insights Into Microbial Ecology, Qiime2, version 2020.11 [39]. The DADA2 program was used for quality-filtering, trimming, de-noising, and merging of sequences [40]. All chimeric sequences were removed through the consensus method in DADA2. Sequence alignment was performed with MAFFT and phylogenetic reconstruction of the aligned sequences occurred in FastTree using the plugins align-

ment and phylogeny [41]. Taxonomic and compositional analyses were conducted with the plugins feature classifier (https://github.com/qiime2/q2-feature-classifier, accessed on 17 December 2021). A pre-trained Naive Bayes classifier based on the Greengenes gg_13_5_otus.tar.tgz Operational Taxonomic Units (OTUs) database (http://greengenes.secondgenome.com/?prefix=downloads/greengenes_database/gg_13_5/, accessed on 17 December 2021/), which had been previously trimmed to the V4 region of 16S rDNA, bound by the 341F/805R primer pair, was applied to paired-end sequence reads to the generate taxonomy tables.

Data generated by Illumina sequencing were uploaded in the NCBI Sequence Read Archive (SRA) and are available under Ac. PRJNA809221.

### 2.7. Alpha Diversity Analysis

Alpha diversity helps us to understand the bacterial community structure in terms of the number of taxonomic groups (richness) and/or the distribution of group abundance (evenness). Richness and evenness of the sites investigated were analyzed through Shannon–Wiener diversity index ($H_{Sh}$) and Gini–Simpson diversity index ($H_{Si}$) [42,43]:

$$H_{Sh} = -\sum_{i=1}^{N_s} p_i \times \ln(p_i) \tag{1}$$

$$H_{Si} = 1 - \sum_{i=1}^{N_s} p_i^2 \tag{2}$$

In these equations, $p_i$ is the ratio between the number of sequences of the OTUs $i$ (as % on the total sequences) and the richness $N_s$ (total number of OTUs identified).

The results of the sites with different $N_s$ were compared by equitability (the even-distributed values) of the indices $H_{Sh}$ and $H_{Si}$ determined as follows:

$$E_{Sh} = \frac{H_{Sh}}{\ln N_s} \tag{3}$$

$$E_{Si} = \frac{H_{Si}}{1 - \frac{1}{N_s}} \tag{4}$$

where $E_{Sh}$ and $E_{Si}$ represent the even-distributed values of $H_{Sh}$ and $H_{Si}$, respectively.

### 2.8. Statistical Analysis

Microbiological count data were subjected to One-Way Variance Analysis (ANOVA) using XLStat software version 7.5.2 for Excel (Addinsoft, New York, NY, USA). Tukey's test was applied only for comparison between different pedons at the same depth. Statistical significance was attributed to $p$ values of $p < 0.05$ and are marked with different letters.

Principal component analysis (PCAn) was carried out to evaluate the relationship between the bacterial phyla and the physicochemical of each sample. Kaiser criterion was used to select the number of principal factors with an eigen value >1.00 [44]. A check of the statistical significance within the data set was performed using Barlett's sphericity test [45]. Data were processed using the XLStat software reported above. In addition, agglomerative hierarchical clustering (AHC) (joining, tree clustering) was performed to group data on soils based on their mutual dissimilarity, measured by Euclidean distances. A cluster aggregation was obtained, applying the single linkage method of Todeschini (1998) [46].

In order to better correlate bacterial taxa and soil properties, a correlation analysis between bacterial phyla and all physicochemical characteristics of the soil samples was performed by means of Pearson's correlation coefficient calculation within the XLStat software reported above. The results were plotted in a heat map.

## 3. Results

### 3.1. Physicochemical Characteristics of Soils

The results of physicochemical analysis of the studied soil plot are reported in Table 1. The concentration of $CaCO_3$ in the control soil Mant 0 (190–200 g/kg) was consistently lower than the values registered in the anthropogenic soil samples Mant 1–Mant 3 (360–640 g/kg). In particular, the last samples showed highly variable data, with the highest levels detected for the deepest aliquots of Mant 1 and Mant 3. On the contrary, clay levels of control soils were higher than those characterizing the anthropogenic soil samples. The EC the values of Mant 0 were lower than values registered for the other pedons and an increasing trend was observed with depth. Regarding $C_{org}$, the values of all anthropogenic pedons were lower than those displayed by the control soil Mant 0. Among anthropogenic sites, the highest value of $C_{org}$ was reached by the sample Mant 2-II (6.97 g/kg), while the highest value in Mant 0 was shown by the most superficial sample Mant 0-I (15.11 g/kg) and decreased with depth until 8.52 g/kg.

**Table 1.** Main physicochemical features of investigated anthropic soils profiles in Giordano area.

| Profile | Depth | $CaCO_3$ Tot (g/kg) | Clay (g/kg) | Silt (g/kg) | Sand (g/kg) | pH ($H_2O$) | EC ($\mu S/cm$) | CEC ($Cmol_{(+)}/kg$) | $C_{org}$ (g/kg) |
|---|---|---|---|---|---|---|---|---|---|
| Mant 0 | I | 200 | 446 | 269 | 285 | 7.83 | 82.0 | 33.75 | 15.11 |
| Mant 0 | II | 190 | 452 | 304 | 244 | 7.89 | 93.2 | 25.00 | 8.72 |
| Mant 0 | II | 200 | 498 | 259 | 243 | 7.76 | 95.5 | 31.25 | 8.52 |
| Mant 1 | I | 550 | 371 | 308 | 321 | 7.87 | 147.6 | 22.50 | 3.10 |
| Mant 1 | II | 640 | 295 | 328 | 377 | 7.90 | 145.5 | 20.00 | 2.13 |
| Mant 1 | III | 640 | 369 | 342 | 289 | 7.89 | 192.5 | 18.75 | 2.90 |
| Mant 2 | I | 550 | 363 | 418 | 219 | 7.87 | 172.5 | 21.30 | 5.03 |
| Mant 2 | II | 360 | 402 | 317 | 281 | 7.88 | 150.0 | 25.00 | 6.97 |
| Mant 2 | III | 420 | 390 | 307 | 303 | 7.90 | 160.5 | 21.30 | 3.87 |
| Mant 3 | I | 460 | 363 | 299 | 338 | 7.86 | 178.9 | 22.50 | 5.62 |
| Mant 3 | II | 640 | 333 | 373 | 294 | 7.88 | 154.9 | 20.00 | 5.23 |
| Mant 3 | III | 640 | 314 | 313 | 373 | 7.88 | 141.1 | 21.30 | 6.00 |

Abbreviations are as follows: $CaCO_3$ Tot, total content of carbonate; EC, electrical conductivity; CEC, cation exchange capability; $C_{org}$, content of organic carbon. Soil samples: Mant 0, control soil (original Vertisol not affected by pedotechnics), while Mant 1, Mant 2 and Mant 3 represent the anthropogenic soils of sites 1 to 3. Depths: I, 0–10 cm; II, 10–30 cm; and III, 30–50 cm.

### 3.2. Analysis of Microbial Communities

The results of the microbiological counts are reported in Table 2. TMA cell densities were in the range 6.06–7.56 CFU per gram of soil. Each pedon displayed a significant decreasing level for TMA with soil depth; TMAn were also at six orders of magnitude per gram of soil and the highest cell densities were registered for the superficial (A) samples. TMAn levels for the samples B and C for all three anthropogenic pedons were consistently lower than the corresponding pedons of the undisturbed soil.

FF and AB groups were characterized by cell densities of 1–3 Log cycles lower than those displayed by TMA. In general, at the highest depth, FF were below the detection limit, with the exception of pedon Mant 2 for which barely 2.0 Log CFU per g dw were found. A higher variability was indicated by AB levels, which were undetectable from B sample for pedon Mant 2, while around 104 CFU per g dw for the other pedons, control soil included.

Bacteria grown on synthetic media CM and NK were isolated to better investigate in future on their role in soil particle aggregation. About 10 colonies sharing the same mucoid appearance and white/yellow color were picked up from agar plates and promptly purified. A total of 834 bacterial cultures presumptively belonging to *Caulobacter* and *Sphingomonas* genera were then microscopically processed, and barely 423 isolates showed a rod-shape.

**Table 2.** Results of plate count agar of the microbial groups.

| Soil | TMA | FF | AB | TMAn | NF | *Sphingomonas* | *Caulobacter* |
|---|---|---|---|---|---|---|---|
| Control soil | | | | | | | |
| Mant 0-I | 6.78 ± 0.10 [b] | 5.74 ± 0.04 [a] | 4.65 ± 0.05 [a] | 6.61 ± 0.11 [a] | 6.90 ± 0.18 [ab] | 6.42 ± 0.06 [c] | 6.87 ± 0.07 [ab] |
| Mant 0-II | 6.34 ± 0.08 [b] | 5.30 ± 0.00 [a] | 4.39 ± 0.09 [a] | 6.19 ± 0.05 [a] | 6.30 ± 0.08 [a] | 6.20 ± 0.10 [a] | 6.52 ± 0.20 [ab] |
| Mant 0-III | 6.14 ± 0.02 [c] | <2 [b] | 4.00 ± 0.00 [b] | 6.12 ± 0.03 [a] | 5.78 ± 0.17 [b] | 6.00 ± 0.08 [b] | 6.31 ± 0.01 [b] |
| Statistical significance | *** | *** | *** | *** | *** | ** | ** |
| Anthropogenic soil | | | | | | | |
| Mant 1-I | 7.00 ± 0.07 [b] | 4.74 ± 0.04 [b] | 4.60 ± 0.20 [a] | 6.45 ± 0.09 [a] | 6.59 ± 0.30 [b] | 6.23 ± 0.03 [d] | 6.64 ± 0.28 [b] |
| Mant 1-II | 6.22 ± 0.01 [bc] | 4.30 ± 0.30 [b] | 4.30 ± 0.20 [a] | 5.71 ± 0.13 [b] | 6.03 ± 0.30 [a] | 5.97 ± 0.15 [b] | 6.20 ± 0.09 [ab] |
| Mant 1-III | 6.15 ± 0.05 [c] | <2 [b] | 4.45 ± 0.15 [a] | 5.75 ± 0.04 [b] | 5.97 ± 0.11 [b] | 5.76 ± 0.14 [c] | 6.29 ± 0.05 [b] |
| Statistical significance | *** | *** | | *** | * | ** | * |
| Anthropogenic soil | | | | | | | |
| Mant 2-I | 7.09 ± 0.09 [ab] | 4.69 ± 0.09 [b] | 5.00 ± 0.25 [a] | 6.52 ± 0.06 [a] | 7.08 ± 0.04 [a] | 6.63 ± 0.1 [b] | 6.78 ± 0.00 [ab] |
| Mant 2-II | 6.06 ± 0.01 [c] | 4.24 ± 0.24 [b] | <2 [b] | 5.43 ± 0.05 [c] | 5.44 ± 0.05 [b] | 5.64 ± 0.00 [c] | 6.5 ± 0.30 [b] |
| Mant 2-III | 6.37 ± 0.06 [b] | 2.00 ± 0.30 [a] | <2 [c] | 5.77 ± 0.01 [b] | 6.00 ± 0.02 [b] | 6.10 ± 0.01 [b] | 6.31 ± 0.08 [b] |
| Statistical significance | *** | *** | *** | *** | *** | *** | * |
| Anthropogenic soil | | | | | | | |
| Mant 3-I | 7.56 ± 0.35 [a] | 5.70 ± 0.00 [a] | 4.59 ± 0.11 [a] | 6.66 ± 0.03 [a] | 7.29 ± 0.04 [a] | 6.81 ± 0.07 [a] | 7.06 ± 0.01 [a] |
| Mant 3-II | 6.77 ± 0.17 [a] | 5.39 ± 0.09 [a] | 4.30 ± 0.25 [a] | 6.06 ± 0.09 [a] | 6.39 ± 0.16 [a] | 6.33 ± 0.03 [a] | 6.59 ± 0.07 [a] |
| Mant 3-III | 6.74 ± 0.00 [a] | <2 [b] | 4.00 ± 0.00 [b] | 6.14 ± 0.02 [a] | 6.43 ± 0.17 [a] | 6.52 ± 0.01 [a] | 6.82 ± 0.13 [a] |
| Statistical significance | ** | *** | * | *** | *** | *** | ** |
| Statistical significance I | ** | *** | | | ** | *** | * |
| Statistical significance II | *** | *** | *** | *** | *** | *** | * |
| Statistical significance III | *** | *** | *** | *** | ** | *** | *** |

Abbreviations are as follows: TMA, total mesophilic count in aerobic condition; FF, filamentous fungi; AB, actinobacteria; TMAn, total mesophilic count in anaerobic condition; NF, nitrogen fixing bacteria. Mant 0, control soil (original Vertisol not affected by pedotechnics), while Mant 1, Mant 2 and Mant 3 represent the anthropogenic soils of sites 1 to 3. Depths: I, 0–10 cm; II, 10–30 cm; and III, 30–50 cm. Results indicate mean values ± S.D. of four microbiological counts (carried out in duplicate for two independent sample collections). Unit of measure: CFU/100 g of dried soil. Soil samples: Data within a column followed by the same letter are not significantly different according to Tukey's test. *. $p \leq 0.05$; **. $p \leq 0.01$; ***. $p \leq 0.001$.

About 40% of the 423 isolates were subjected to RAPD analysis to differentiate the cultures at strain level, and 63 different profiles (Figure 3) indicated a certain biodiversity within putative EPS-producing bacteria. The bacteria showing different RAPD patterns were considered distinct strains. The dendrogram showed that the majority of strains grouped according to their genus and species, even though the two species of *Stenotrophomonas*, St. *indicatrix* and St. *rhizophila*, clustered quite distantly, the first within *Variovorax* and the second within *Pseudomonas* and *Lysobacter*. All strains were further processed by 16S rRNA gene sequencing and a remarkable species diversity emerged from these results. The isolates were allotted into three phyla (Firmicutes, Actinobacteria and Protobacteria) consisting of 19 genera; the most numerous bacterial groups were represented by *Bacillus* and *Pseudomonas*, which were found in all pedons analyzed. Surprisingly, none of the 63 strains were identified as *Sphingomonas* or *Caulobacter*.

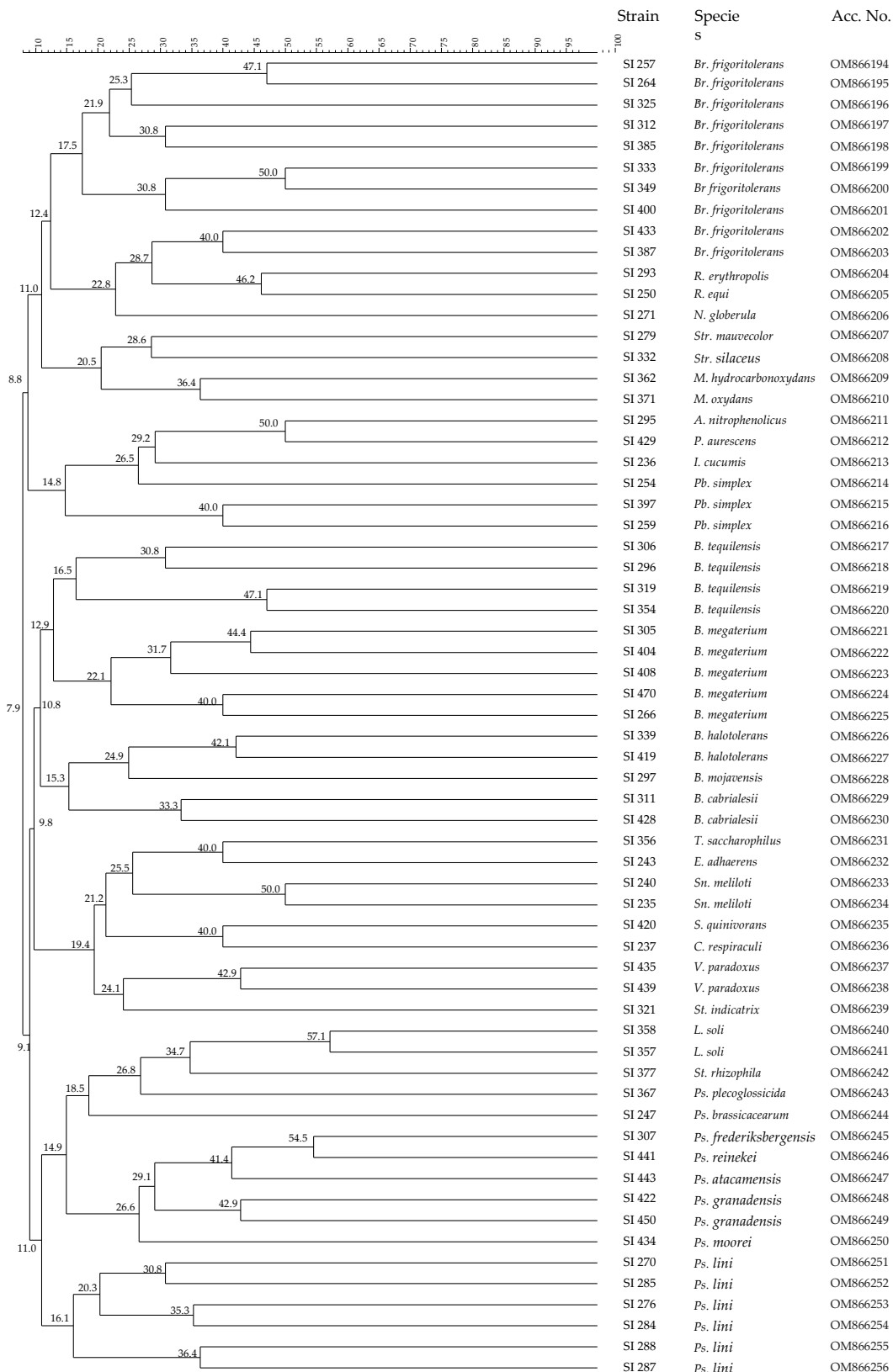

**Figure 3.** Dendrogram obtained from combined RAPD-PCR patterns generated with three primers (M13, AB106 and AB111) of the dominant bacteria identified from anthropogenic soils. Abbreviations: A., *Arthrobacter*; B., *Bacillus*; Br., *Brevibacterium*; C., *Cupriavidus*; E., *Ensifer*; I., *Isoptericola*; L., *Lysobacter*; M., *Microbacterium*; N., *Nocardia*; P., *Paenarthrobacter*; Pb., *Peribacillus*; Ps., *Pseudomonas*; R., *Rhodococcus*; S., *Serratia*; Sn., *Sinorhizobium*; St., *Stenotrophomonas*; Str., *Streptomyces*; T., *Terribacillus*; V., *Variovorax*.

### 3.3. Culture-Independent Analysis

In order to deeply analyze the bacterial communities of control and anthropogenic soil samples, a next-generation sequencing (NGS) approach was also applied. This approach was necessary because the great majority of microorganisms (>99%) of natural ecosystems cannot be detected by means of the culture-dependent tools alone [47]. In this study, all soil samples were investigated using MiSeq Illumina technology, and the results are reported in Figure 4 (at phylum level) and Figure S1 (all taxonomic levels mixed). The total bacterial diversity of the soil under evaluation was composed of 11 phyla—Actinobacteria, Chloroflexi, Firmicutes and Proteobacteria were present in all pedons at the three depths investigated.

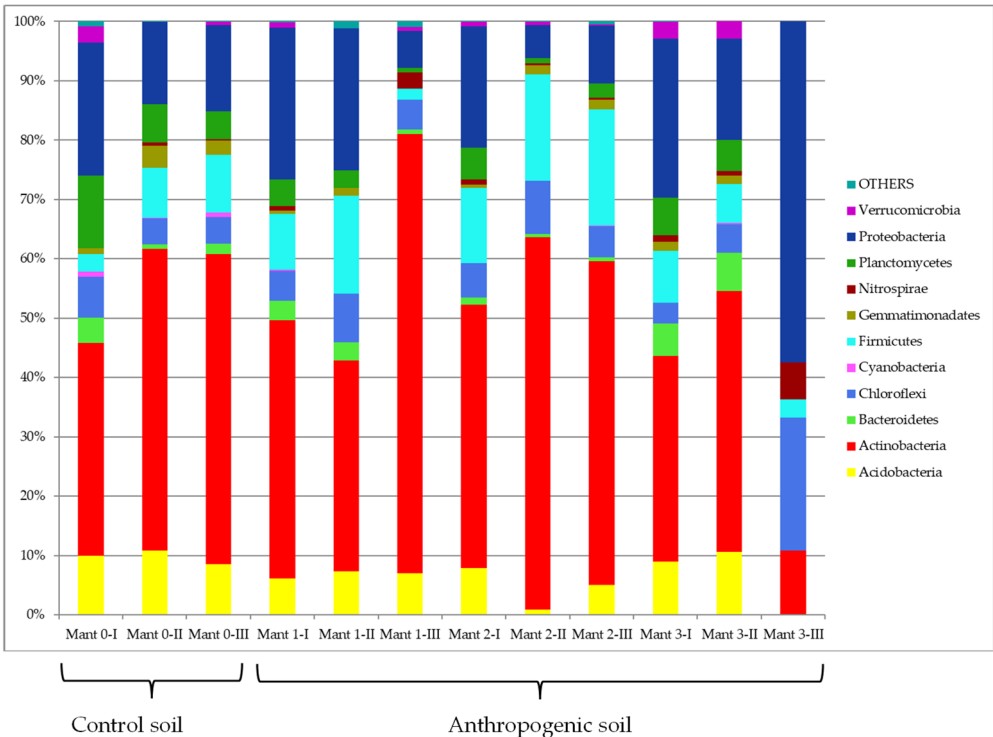

**Figure 4.** Relative abundances (%) of bacterial phyla identified by MiSeq Illumina in soil samples. Abbreviations are as follows: Soil samples: Mant 0, control soil (original Vertisol not affected by pedotechnics), while Mant 1, Mant 2 and Mant 3 represent the anthropogenic soils of sites 1 to 3. Depths: I, 0–10 cm; II, 10–30 cm; and III, 30–50 cm.

With the exception of pedon Mant 3-III, Actinobacteria represented the most abundant phylum in almost all samples analyzed, reaching the maximum level (74.04%) in Mant 1-III. Among this phylum, only members of Gaiellaceae family were found in all samples, with percentages ranging from 4.46% in Mant 3-I and 33.06% in Mant 1-III. Within this phylum, radiotolerant and halotolerant bacteria belonging to the *Rubrobacter* genus were quite ubiquitous, even though their presence was particularly variable (0.5–23.5%).

The presence of Firmicutes, especially within the genus *Bacillus*, was detected in almost all pedons of both control and anthropogenic soils; in particular, this genus was detected at very high percentages in all Mant 2 pedons (12.63–19.06%). Additionally, the order of Clostridiales was found in almost all pedons, reaching the maximum abundance (9.08%) in Mant 1-II.

Six families of Proteobacteria (Bradyrhizobiaceae, Rhodospirillaceae, Syntrophobacteraceae, Enterobacteriaceae, Moraxellaceae and Pseudomonadaceae) were also ubiquitous in the soil samples, but at percentages lower than those of Actinobacteria. Only in Mant 3-III did Proteobacteria account for 57.45% of total OTUs. *Erwinia* and *Acinetobacter* were the most consistent genera, with 18.34 and 16.81%, respectively.

Two main classes within Chloroflexi phylum were present in our samples, in particular, Anaerolineae and Ktedonobacteraceae. The first group was detected in all soil samples, with percentages ranging from 0.03% in Mant 3-I to 8.40% in Mant 3-III, while the second was found only in Mant 3-III, but the relative abundance (14.01%) was quite high.

### 3.4. Analysis of Biodiversity Indexes (Alpha Diversity)

The intra-site diversity and richness were assessed by alpha diversity considering the phyla of the bacterial community. The distribution of phyla among soils samples was studied and the results are reported in Table 3.

**Table 3.** Quantitative data analysis of bacterial phyla (alpha diversity).

| Samples | No. of Phyla | $H_{Sh}$ | $E_{Sh}$ | $H_{Si}$ | $E_{Si}$ |
|---|---|---|---|---|---|
| Mant 0-I | 11 | 1.84 | 0.74 | 0.79 | 0.86 |
| Mant 0-II | 11 | 1.58 | 0.64 | 0.70 | 0.76 |
| Mant 0-III | 12 | 1.58 | 0.64 | 0.68 | 0.75 |
| Mant 1-I | 12 | 1.64 | 0.66 | 0.73 | 0.79 |
| Mant 1-II | 9 | 1.73 | 0.69 | 0.78 | 0.85 |
| Mant 1-III | 10 | 1.06 | 0.42 | 0.44 | 0.48 |
| Mant 2-I | 12 | 1.64 | 0.66 | 0.73 | 0.80 |
| Mant 2-II | 11 | 1.20 | 0.48 | 0.56 | 0.61 |
| Mant 2-III | 12 | 1.44 | 0.58 | 0.65 | 0.71 |
| Mant 3-I | 11 | 1.82 | 0.73 | 0.78 | 0.85 |
| Mant 3-II | 11 | 1.77 | 0.71 | 0.75 | 0.82 |
| Mant 3-III | 5 | 1.17 | 0.47 | 0.60 | 0.66 |
| Reference value for a perfect even community | | 2.48 | 1 | 0.92 | 1 |

Abbreviations: $H_{Sh}$, Shannon–Wiener diversity index; $E_{Sh}$, equitability of $H_{Sh}$; *his*, Gini–Simpson diversity index; $E_{Si}$, equitability of $H_{Si}$; Soil samples: Mant 0, control soil (original Vertisol not affected by pedotechnics), while Mant 1, Mant 2 and Mant 3 represent the anthropogenic soils of sites 1 to 3. Depths: I, 0–10 cm; II, 10–30 cm; and III, 30–50 cm.

The total number of phyla (n. = 12) was found only in four samples, while only one soil sample (Mant 3-III) was characterized by a very low number of phyla (n. = 5). The samples collected at the highest depths showed the lowest biodiversity since the samples Mant 1-III, Mant 3-III and Mant 2-II were characterized by a $H_{Sh}$ of 1.06, 1.17 and 1.20, respectively. Shannon indexes increased until 1.82 and 1.84 in samples Mant 3-I and Mant 0-I, respectively, indicating that the most superficial soil horizon is characterized by a high bacterial biodiversity. This trend was also confirmed by Simpson's index, since the lowest $H_{Si}$ (0.44) was registered for the sample Mant 1-III and the highest (0.79) for sample Mant 0-I. Regarding this last biodiversity index, except at site Mant 2, the samples collected at the first two depths (0–10 cm and 10–30 cm) from the anthropogenic plot were highly comparable. Both biodiversity indexes and their equitability showed that the bacterial community of control and anthropogenic soils are not particularly different.

### 3.5. Multivariate Data Analysis

All pedons were subjected to AHC (Figure 5) to obtain their classification in accordance with their mutual dissimilarity and relationship based on physicochemical parameters and relative abundance (%) of microbiota. Anthropogenic soil samples clustered almost randomly considering their depth. In particular, the only two samples that showed a very low dissimilarity were Mant 1-I and Mant 2-I, while Mant 3-I clustered with Mant 3-II, and Mant 2-II with Mant 2-III. Considering a dissimilarity level of 30%, all Mant 0 samples formed a single mega-cluster distant from the other clusters. The results showed that the mega-cluster of Mant 0 samples is significantly different from the other two clusters of anthropogenic soils with values of dissimilarity higher than 60.65% and values of within-

class variance lower than 13.20%. This analysis clearly demonstrated high dissimilarity among anthropogenic soil samples.

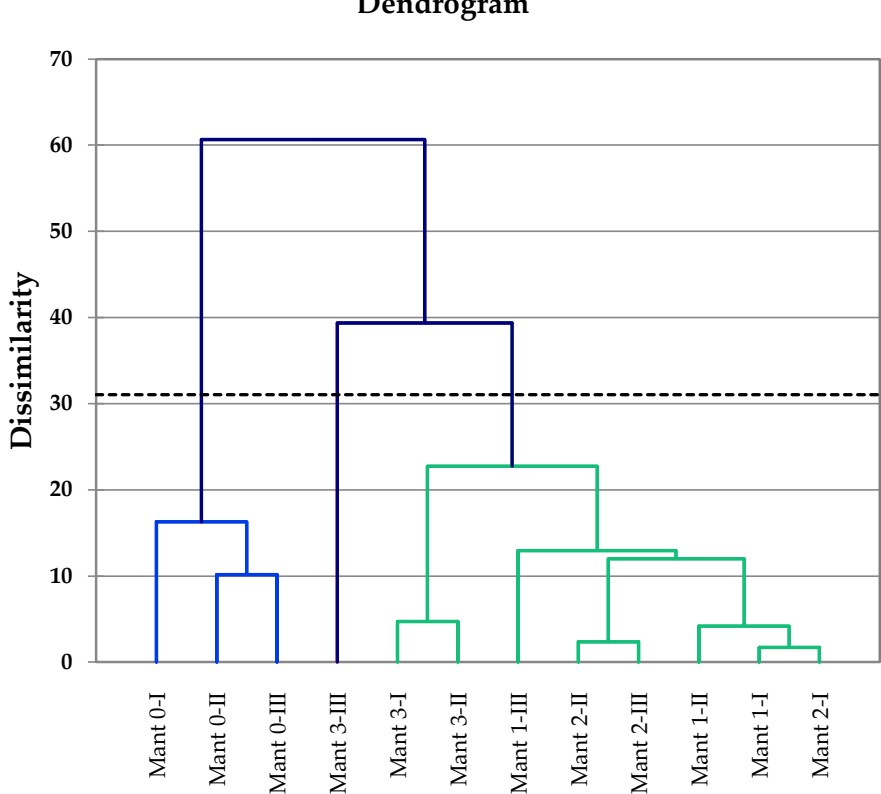

**Figure 5.** Dendrogram obtained from the hierarchical cluster analysis (AHC). Abbreviations are as follows: Soil samples: Mant 0, control soil (original Vertisol not affected by pedotechnics), while Mant 1, Mant 2 and Mant 3 represent the anthropogenic soils of sites 1 to 3. Depths: I, 0–10 cm; II, 10–30 cm; and III, 30–50 cm.

Furthermore, PCA was also carried out with the aim to discriminate a very high number of physicochemical and microbiological parameters in a few factors (Figure 6). Factors 1 and 2 had an eigen value higher than 3 and explained 58.71% of total variability (38.56% and 20.14%. for F1 and F2., respectively). Biplot representation shows that CEC and $C_{org}$ strongly influence the F1. Considering microbial groups, Acidobacteria, Planctomycetes and Cyanobacteria showed a strong influence on the factor F1, while Verrucomicrobia and Bacteroidetes only showed a weak influence on F1, and this contributed to grouping control soil samples into a mega cluster. On the contrary, Proteobacteria and Chloroflexi strongly influence the F2 factor.

To examine the reciprocal influence of bacterial community and soil characteristics, a correlation analysis was performed between bacterial phyla and physicochemical parameters of the samples investigated (Figure 7). The most significant positive correlations of Planctomyces, Gemmatimonadates, Cyanobacteria and Acidobacteria were found for clay, CSC and $C_{org}$. A highly positive correlation with clay was also showed by Actinobacteria, with $C_{org}$ by Bacteroidetes, and with CSC and $C_{org}$ by Verrucomicrobia. Chloroflexi, Nitrospirae, Proteobacteria. Non-identified members of the community showed the highest positive influence on limestone and sand. A positive influence on the last characteristic was also registered for Bacteroidetes. Actinobacteria and Firmicutes were mainly impacted soil conductivity. Finally, Firmicutes and other members influenced soil pH.

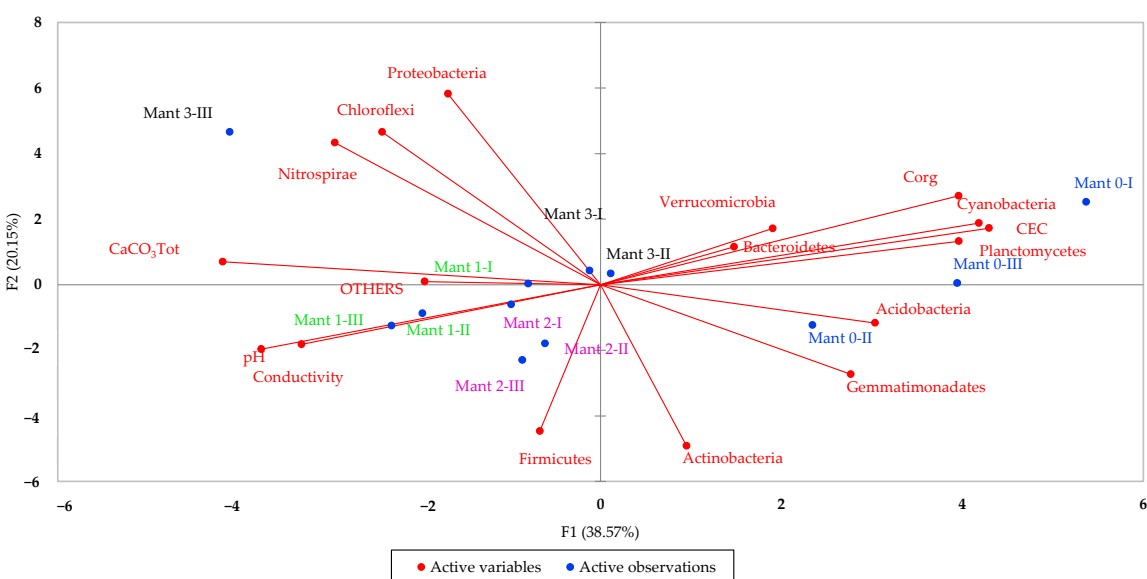

**Figure 6.** Principal component analysis (PCA) among the microbiological and physicochemical characteristics of soil sample. Abbreviations are as follows: CaCO$_3$ Tot; total content of carbonate; EC, electrical conductivity; CEC, cation exchange capability; C$_{org}$, content of organic carbon. Soil samples: Mant 0, control soil (original Vertisol not affected by pedotechnics), while Mant 1, Mant 2 and Mant 3 represent the anthropogenic soils of sites 1 to 3. Depths: I, 0–10 cm; II, 10–30 cm; and III, 30–50 cm.

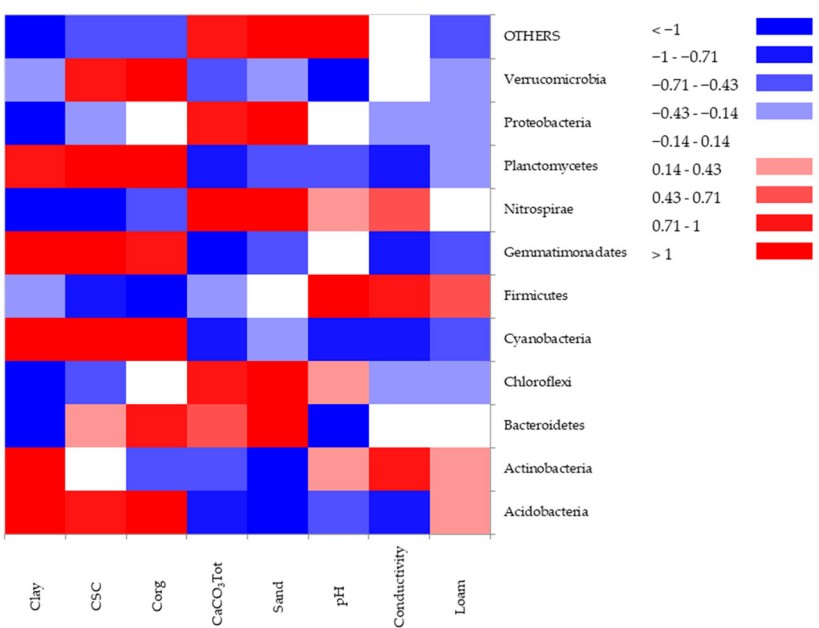

**Figure 7.** Pearson's correlation between bacterial phyla and physicochemical characteristics of soil samples. Colour intensity indicates the level of association. Abbreviations are as follows: CaCO$_3$ Tot; total content of carbonate; EC, electrical conductivity; CEC, cation exchange capability; C$_{org}$, content of organic carbon.

## 4. Discussion

The success of anthropogenic soils is mainly due to the high economic incomes deriving from high yields through high labor inputs [48]. Since the 1970s, the entire agricultural area of Palma di Montechiaro (Sicily) has been subjected to a change in land use for

economic purposes. Olive yards and almond yards were converted into vineyards [49]. Moreover, the cultivated soils of this area were transformed by native farmers into anthropogenic soils, adding consistent amounts of calcareous marls in order to improve the organoleptic traits of table grape.

The chemical parameters of the anthropogenic soils under investigation were compared to those of control soil. The concentration of $CaCO_3$ of the control soil was between 190 and 200 g/kg, comparable to that characterizing other Haploxerert soils located in the same area [50]. The concentration of calcium carbonate of the anthropogenic soils was higher, in the range 420–640 g/kg. These high results have been also registered for soils from Quaternary limestone [51]. The values of pH of anthropogenic soils were around 7.9, an average value displayed by soils with calcareous marls [6,50–52]. These high pH values are due to the high concentrations of $CaCO_3$ that exerts a buffer capacity [53]. Furthermore, CEC of control soil was different from that of the anthropogenic soils analyzed. Regarding this parameter, the average value of the anthropogenic soils was lower than that of control soil because of the lower clay percentage after intervention [51,54–57]. The concentration of $C_{org}$ Tot was highest in control soil; this parameter ranged between 8.52 and 15.11 g/kg, commonly registered for Vertisols [50]. The level of $C_{org}$ Tot in anthropogenic soils was consistently low (2.13 to 6.97 g/kg) and this is due to the dilution effect determined by the addition of calcareous-marl material.

To our knowledge, anthropogenic soil microbiology in Sicily is almost unexplored, and the basic relationships among soil modifications and bacterial evolution merit further investigation. In general, the anthropic actions applied to soil might cause microbial biodiversity reduction in the short term [58]. Thus, in this work, microbial imaging analysis was performed after 7 years from soil modification, just before the first cycle of plant cultivation. The microbial communities of the soils located in Palma di Mantechiaro were first investigated by a polyphasic culture-dependent approach to provide direct evidence on the viable populations. Plate count analysis carried out on soil samples showed that TMA levels were, on average, in the range 6.7–7.5 Log CFU per g dw. These levels characterize commonly uncultivated soils [59,60]. In general, TMA decreased with depth and this finding is imputed to nutrient and oxygen limitations [61]. FF levels were 1 or 2 orders of magnitude lower than those of TMA. These results are due to the fact that bacteria fill an ecological niche composed of aerobic, facultative anaerobic and N-fixer microorganisms [56], which is much larger than that occupied by fungi. The same levels of microscopic fungi were reported by numerous authors in the superficial parts of bulks soils [60,61]. Generally, in agricultural soils, FF decrease at undetectable levels already at 20–30 cm, but in our anthropogenic soil samples, the absence of fungi was only recorded at a 30–50 cm depth. This could be explained by a greater penetration of oxygen into the deeper layers of soils enriched with calcareous material. In all cases, the viable levels of the microbial groups investigated decreased with depth. For Actinomycetes, the count range was close to that of FF, settling on values ranging between 4.0 and 5.0 Log CFU per g dw, confirming what has been observed by other authors [62,63]. The same authors also reported that the highest Actinomycetes densities are mostly found in the superficial soil horizon.

In order to identify bacteria playing a direct role in soil aggregate formation, the genera more typically associated with EPS production were investigated after colony isolation. Bacteria grown on media used to count *Sphingomonas* and *Caulobacter* were purified and cultured in broths in order to perform 16S rRNA gene sequencing. Despite growth on CM and NK, none of the 63 presumptive *Sphingomonas* and *Caulobacter* isolates were confirmed to belong to these genera. These findings are imputable to the scarce selectivity of the two agar media used for these genera. The isolates were confirmed to represent 63 different strains by RAPD investigation. The comparison of the 16S rRNA gene sequences with those available in GenBank/DDBJ/EMBL and EZtaxon databases allotted the 63 strains into 19 genera of three phyla: Actinobacteria, Firmicutes and Protobacteria. All three phyla generally represent the most abundant soil bacterial groups [64,65]. Bacteria of the genus

*Bacillus* have been found in all pedons. Five strains were identified as *Bacillus megaterium,* which is known for its ability to produce EPS [66]. Furthermore, eight *Pseudomonas* strains have been also identified. Bacteria of this genus are widely known to produce EPS [16]. They have been used singly or in association with Stenotrophomonas to stabilize aggregates of artificial soils [67]. Among presumptive EPS-producing bacteria, some strains were identified as *Cupriavidus* and *Sinorhizobium*, both N-fixing bacteria. Besides *Bacillus* and *Pseudomonas*, some of the other bacteria identified, such as *Streptomyces* and *Serratia,* are considered plant-growth-promoting bacteria [68].

In order to deepen the information about the bacterial community, MiSeq Illumina technology was applied to total bacterial DNA extracted from all soil samples. The superficial layer of control soil was mainly characterized by the presence of Actinobacteria, Acidobacteria, Protobacteria and Planctomycetes. Apart from Planctomycetes, the other bacterial orders are generally detected in cultivated and uncultivated soils [69,70]. The dominant phyla of bacteria for the anthropogenic soils were almost comparable to those of control soil. However, some exceptions were observed; Mant 3 was characterized by Proteobacteria, Actinobacteria and Chloroflexi as the bacteria found at the highest percentages. These groups are generally detected in soils subjected to conventional and minimum tillage [71]. The high relative abundance of Ktedonobacteraceae in Mant 3-III is unsurprising, since this family uses sugars and peptides as carbon sources [72]. Among these bacteria, Acidobacteria are known as ecological indicators because they are oligotrophic and are present in soils with a scarce presence of nutrients, while, on the contrary, being detected at very low levels in fertilized soils [73]. On the other hand, Proteobacteria are considered copiotrophs [74] and are generally commonly detected in environments with greater nutritional opportunities [75]. In the current study, Proteobacteria was widely represented, for example in soil sample Mant 3-III, where their presence accounted for 57.45% of relative abundance. The dominance of this phylum over the bacterial community of soil has been reported by several authors [76–79]. These bacteria play important roles in soil; for example, Bradyrhizobiaceae and Rhodospirillaceae found almost in all samples are N-fixers [80,81], while Oxalobacteraceae, found in A and B layers of control and anthropogenic soils, are the main degraders of soil organic matter and they are also involved in production of phytohormones, such as auxin, gibberellin and siderophores [82]. These families of Protobacteria are also indicators of soil fatigue [70]. Copiotrophs consistently contribute to carbon mineralization [83].

Actinobacteria and Firmicutes cannot be considered copiotrophic nor oligotrophic [59], but rather as ubiquitous bacteria capable of living in completely different conditions such as forest and grassland soils [78,79,84] and even in polluted soils [77,85,86]. Specifically, *Microbacterium* detected in almost all soils are able to degrade polycyclic aromatic hydrocarbons with a high molecular weight [87]. In particular, the culture-dependent approach confirmed that *Microbacterium oxydans* and *Microbacterium hydrocarbonoxydans* were present in viable form in the superficial horizon.

An abundance of Chloroflexi are identified among soil bacterial communities [88–91]. The primary reason for their ubiquity is due to their metabolic diversity; this phylum includes heterotrophs, lithotrophs and phototrophs adapted to both oxic and anoxic environments [92]. All soil samples analyzed in this study displayed bacteria within this phylum in a range similar to those previously documented by other authors, even though, surprisingly, Mant 3-III showed 22.42% of relative abundance allotted into Chloroflexi. With regard to *Sinorhizobium*, *Pseudomonas*, *Bacillus* and *Lysobacter* genera, the culture-independent approach showed some inconsistencies with the culture-dependent approach, since their sequences were not amplified from DNAs extracted from samples that showed their presence in viable form. This is not surprising, since these genera might have been allotted into unassigned OTUs or their sequences were damaged by nucleases or were below 0.1% abundance and not considered during classification.

Diversity indexes are useful to evaluate differences among environmental samples [93]. In this study, Shannon and Simpson indexes were calculated to compare control and an-

thropogenic soils at the depths 0–10, 10–30 and 30–50 cm in terms of bacterial taxa diversity at the phylum level. Due to the dominance of given phyla, the bacterial community was basically uneven in all samples investigated. Alpha diversity analysis clearly showed that bacterial diversity decreases with depth, and that among anthropogenic and control soils, the differences were quite negligible.

Finally, all data were subjected to a multivariate statistical analysis. Both PCA and AHC have been largely used to study the behavior of microorganisms in soil, the influence of different external variables in the distribution of microorganisms or to study different anthropogenic soils [94,95]. The main chemical parameters implicated on microbial diversity were pH and $C_{org}$. These two parameters are generally used as predictors of bacterial diversity in a huge number of sites around the world [96,97]. In those studies, a soil pH around 7 correlated with a high biodiversity, while a decrease in pH determined a reduction in bacterial biodiversity. Regarding the main bacterial groups found in our study, Actinobacteria were not significantly correlated with pH and $C_{org}$, and this could be explained by the ubiquity and capacity of these bacteria to live in different terrestrial environments, even the most extreme soils characterized by nutrient deficiencies [98–100]. Furthermore, Proteobacteria, Chloroflexi and Firmicutes were negatively correlated with $C_{org}$ value. Regarding the direct influence of bacterial taxa on soil characteristics, the correlation analysis confirmed the trend found for PCA. The spatial position of the three pedons could not be a factor driving any variance in the microbial profile, since the only factor dominant and regulating all soil variables is the man action in generating such soils by transformation of the original Vertisol using the described pedotechnique. The effects of any natural soil genetic factors were canceled at the moment of the creation of the anthropogenic soil.

## 5. Conclusions

In conclusion, plate count results showed a significant decrease in almost all microbial groups with increasing sample depth. The genetic analysis of the strains highlighted the presence of 19 different genera, denoting a remarkable biodiversity within the plot analyzed. Only two genera (*Bacillus* and *Pseudomonas*) were found in all pedons. The results obtained from the isolation and sequencing of putative EPS-producing bacteria identified the presence of *Bacillus*, *Pseudomonas* and *Stenotrophomonas*. Further studies will be performed to test their potential to improve the aggregation of soils in a Mediterranean environment. Next-generation sequencing indicated that the most represented phyla in anthropogenic soils are Actinobacteria and Proteobacteria. Viable dominant populations indicated that the anthropogenic soil hosted a mature microbial community useful for land cultivation.

**Supplementary Materials:** The following supporting information can be downloaded at: https://www.mdpi.com/article/10.3390/land11050748/s1, Figure S1: Relative abundances (%) of bacteria identified by MiSeq Illumina in soil samples.

**Author Contributions:** Investigation, P.B., E.F. and R.D.G.; data curation, P.B. and R.D.G.; visualization, P.B. and G.L.P.; writing—original draft preparation, P.B.; conceptualization, C.D. and L.S.; resources, C.D.; supervision, C.D.; writing—review and editing, L.S. and G.L.P.; funding acquisition, L.S.; project administration, G.L.P. All authors have read and agreed to the published version of the manuscript.

**Funding:** This research has received co-funding from the European Commission's ERASMUS+ Programme under grant agreement No 2017-1-SE01-KA203-034570.

**Institutional Review Board Statement:** Not applicable.

**Informed Consent Statement:** Not applicable.

**Conflicts of Interest:** The authors declare no conflict of interest.

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
