# Peer review of "Microbiological Analysis and Metagenomic Profiling of the Bacterial Community of an Anthropogenic Soil Modified from Typic Haploxererts"

_land, doi:10.3390/land11050748_

Round 1
Reviewer 1 Report
This work is interesting. To isolate bacteria actively involved in soil particle aggregation, colonies with mucoid appearance were differentiated at strain level and genetically identified: the major groups were represented by Bacillus and Pseudomonas. MiSeq Illumina analysis identified actinobacteria and Firmicutes as main groups.
Does the isolates with EPS also were found in metagenomic data ? please disscuss this question in the paper.
Author Response
Answers to Reviewer 1:
This work is interesting. To isolate bacteria actively involved in soil particle aggregation, colonies with mucoid appearance were differentiated at strain level and genetically identified: the major groups were represented by Bacillus and Pseudomonas. MiSeq Illumina analysis identified actinobacteria and Firmicutes as main groups.
- Thanks for your comments. All your suggestions were considered in this response letter.
Does the isolates with EPS also were found in metagenomic data ? please disscuss this question in the paper.
- Thanks a lot for this interesting question. Metagenomic analysis is a powerful tool to analyse data. However, the identification of EPS producers by metagenomics necessitate a different approach with primers specific for EPS gene clusters. This approach is able to identify all producers but does not provide the certainty that they are cultivable. We prefer to apply a classical approach based on the isolation of mucoid colonies (performed in the present study) followed by the characterization of the EPS (to be done in future).
We would like to report one of the strategy to provide answer to your question: “The microbial communities of the sites could be investigated for their EPS production by culture-independent next generation 16S rRNA gene based sequence analysis to verify the presence of species/genus implicated in EPS production. If so, a whole metagenome shotgun (WMGS) sequencing could be analysed to retrieve insights into the genomic composition and arrangement of complex microbial consortia. A combination of in silico analyses based on nucleotide and protein sequences will facilitate the identification of genetic material belonging to EPS producers.”. However, a direct discussion of this point in the text can create expectations about this approach in the present manuscript by the readers, but WMGS has not been performed. For this reason, we would prefer not to insert any sentence on this point in the Discussion, but we are opened to further suggestions regarding how to report this information in the revised version of the paper.

Reviewer 2 Report
The authors provide a well designed and well-written study on the influence of Antrhosols on bacterial microbial communities. While the outline is clear and the results and discussion is sound, the paper lacks depth in the microbial community analysis.
Introduction: Very well written.
Methodology: Sampling and culture-dependent and culture-independent analysis are well described and give sufficient detail. Sound methodology of primer selection and bioinformatics in the analysis. The statistical analysis could be more detailed. It is however a pity that the fungal community was not investigated.
The discussion is clearly written and addresses the outcomes of the results. What is missing, however, is a relation of the taxa in the bacterial community analysis with the chemical parameters. It would give definite value to the paper if chemical analyses correlate with certain taxa, this is especially important as e.g. Mant 3 has high variability between samples compared e.g. to Mant 1.
Major points:
While the study is well designed, the microbial community analysis lacks depth. Especially data on soil-type diversity is missing (Alpha-diversity). The authors are further encouraged to perform in addition a taxa specific group comparison between parameters and a correlation analysis between taxa and phyio-chemical soil properties. This e.g. could be done by the Rhea pipeline when using R or another statistical tool
Minor points:
Figure 4: The figure is difficult to read. It would be helpful if the four soils could be 1) better distinguished from each other and 2) there are too many taxa listed to be distinguishable and 3) there are different taxonomic levels mixed.
Figure 6: Please use different colours for soil types, also the no. of repetition does not matter here (A, B, C can be omitted). Please also adjust the legend - active variables / active observations. The chemical paramteres / taxa are both related to soil types jointly, which is not very useful as these are only pure correlations with limited inhibition of confounding.
Author Response
Answers to Reviewer 2:
The authors provide a well designed and well-written study on the influence of Antrhosols on bacterial microbial communities. While the outline is clear and the results and discussion is sound, the paper lacks depth in the microbial community analysis.
- Thanks for your comments. All your suggestions were considered and the changes in the text were highlighted in green.
Introduction: Very well written.
- Thanks for your positive response.
Methodology: Sampling and culture-dependent and culture-independent analysis are well described and give sufficient detail. Sound methodology of primer selection and bioinformatics in the analysis. The statistical analysis could be more detailed. It is however a pity that the fungal community was not investigated.
- More details have been provided in paragraph 2.8 regarding statistical treatment of data (L420-427).
Regarding fungal community, we agree with the reviewer that data on fungi would be interesting. However, we work in a laboratory that focuses exclusively on bacteriology. Fungal microorganisms, especially moulds, are bad contaminants for a microbiology lab that works on bacteria. Under our laboratory conditions, we routinely perform plate counts of filamentous fungi, but we are not supposed to open Petri dishes after fungal development to avoid environmental contaminations. This is the reason why we included the counts of fungi, but we did not characterize them furtherly. However, analyses of fungi will be performed by our mycologist colleagues of the same department and a PhD student will focus his research on this aspect in the next future. We reflected on this comment and we actually created expectations about the fungal community due to the too general title of the paper. In order to better encounter the comment of the reviewer, the title has been modified to be more specific for the content of the paper in “Microbiological analysis and metagenomic profiling of the bacterial community of an anthropogenic soil modified from Typic Haploxererts” (L2-4).
The discussion is clearly written and addresses the outcomes of the results. What is missing, however, is a relation of the taxa in the bacterial community analysis with the chemical parameters. It would give definite value to the paper if chemical analyses correlate with certain taxa, this is especially important as e.g. Mant 3 has high variability between samples compared e.g. to Mant 1.
- We put more attention of this aspect during manuscript revision. Please see the answer to the major point.
Major points:
While the study is well designed, the microbial community analysis lacks depth. Especially data on soil-type diversity is missing (Alpha-diversity). The authors are further encouraged to perform in addition a taxa specific group comparison between parameters and a correlation analysis between taxa and phyio-chemical soil properties. This e.g. could be done by the Rhea pipeline when using R or another statistical tool
- Alpha-diversity analysis was added to the text (L321-331,621-641,977-983). A new table (Table 3) showing diversity parameters was provided in the main text and new references were added (L1165-1168,1265-1266).
We positively accepted the suggestion to correlate the bacterial taxa with the physicochemical parameters of soil. To this purpose, the OTUs identified by Illumina at phylum level were correlated with the physicochemical parameters by the Pearson’s correlation coefficient of XL-Stat and plotted in a heat map. We hope this correlation encounters the reviewer expectations. The text was modified consequently (L428-431,725-744, 996-997) and a new figure (Fig. 7) was added.
Minor points:
Figure 4: The figure is difficult to read. It would be helpful if the four soils could be 1) better distinguished from each other and 2) there are too many taxa listed to be distinguishable and 3) there are different taxonomic levels mixed.
- Fig. 4 was modified for clarity as suggested. The different soils were better indicated in the x-axe of the figure. The number of taxa was reduced considering only the different phyla. Thus, 5 of relative abundance are referred to OUT’s grouped per phylum. However, we think that the old figure was very informative especially regarding the different levels identified for the OUT’s, for this reason we kept it in the manuscript, but as supplementary material (Fig. 1S).
Figure 6: Please use different colours for soil types, also the no. of repetition does not matter here (A, B, C can be omitted). Please also adjust the legend - active variables / active observations. The chemical paramteres / taxa are both related to soil types jointly, which is not very useful as these are only pure correlations with limited inhibition of confounding.
- Fig. 6 was modified for clarity as suggested. Different colours were used for the different soil type. The letter A, B and C were replaced by the Roman numbers I, II and III. Please consider that they refer to different depths, they are not replicates. Data used were average data of the technical replications at each depths.

Reviewer 3 Report
The anthropic actions applied to soil might cause a microbial biodiversity reduction in the short term. The present work was carried out to assess the soil microbial community and to investigate specifically on extracellular polymeric substances (EPS) producing populations of an anthropogenic soil prior to its first cultivation cycle. These substances positively affect the structure, porosity, fertility and productivity of the soil systems. The microbial imaging analysis was performed after 7 years from soil modification. It was shown that the anthropogenic soil hosted a mature microbial community useful for land cultivation. Further studies will be performed to test their potential to improve the aggregation of soils in Mediterranean environment.
Essentially, there are no questions to the article. You may wish to pay attention to the following points:
1. Section 3.4 presents data of statistical processing of data on physicochemical parameters and relative abundance of microbiota. In particular, Figure 5 shows a dendrogram obtained as a result of a hierarchical cluster analysis of soil samples, and Figure 6 shows, if not trends, then some similarity of the grouping of figurative points of soil samples on the principal component diagram. These data are not discussed. For example, Mant 3 is different from Mant 1 and Mant 2, while being closer to Fiume Palma Stream (I could be wrong) and probably lower. It is also possible that the authors believe that the observed differences are not significant or these nuances go away from the topic of the article.
2. The use of "marly limestone", in my opinion, is not correct. Based on the composition, marl is a mixture of clay and carbonate (most often calcite). That is, simplistically, marly limestone = mixture of (limestone + mudstone) and limestone, and if you open the brackets, then it is either calcareous marlstone or argillaceous limestone. Perhaps from the point of view of structure or facies, or traditional names of fertilizers, the use of "marly limestone" will seem convenient to you, but at the end of this path, the terminology will cease to be universal.
Specific remarks:
Lines 14-15. If numerical values of calcium carbonate content in soil are given for a control soil, the corresponding numerical value for anthropogenic soil should probably be given.
Line 33. "Anthrosol".
Line 36. focused.
Lines 84-85. "The lithology dates back to the Pliocene and Miocene consisting of <sediments type1>, <sediments type2> and sediments." There is an error in this sentence. If it makes no sense to describe the lithology in more detail, then at the end of the sentence "and sediments" can be replaced by "and other sediments".
Line 85. marly gypsum - Perhaps you mean gypsum marl?
Line 98. You might want to change the phrase so that the word "depth" doesn't appear twice.
Line 107. If there is a need to emphasize the collection of each pedon in duplicate, then a brief explanation should probably be given. For example, one sample was selected for physical and chemical studies, and the second sample was selected for microbiological studies. If it could be assumed that two specimens were selected to increase the volume of the pedon sample, then the question of the representativeness of the samples, more precisely, the volume of samples and the homogeneity of the substance in terms of the studied parameters, would be relevant.
Lines 107-108. If there was a reason why soil samples affected and not affected by pedotechnics were taken specifically at these same depths - 0-10, 10-30, 30-50 cm, then perhaps it would be appropriate to devote a couple of lines to its description here (or provide a link to the description of the method). Perhaps it should be clarified that the A B C horizons do not reflect the structure of the soil profile?
Line 116. a portable conductivity-meter cond 7 - May be Cond 7 or COND 7? If you do not want to write the name of the device with a capital letter, then perhaps you would prefer to write it in quotation marks?
Line 269. Space before Results.
Line 388. pH.
Line 526. Typo - extra number 5.

Author Response
Answers to Reviewer 3:
The anthropic actions applied to soil might cause a microbial biodiversity reduction in the short term. The present work was carried out to assess the soil microbial community and to investigate specifically on extracellular polymeric substances (EPS) producing populations of an anthropogenic soil prior to its first cultivation cycle. These substances positively affect the structure, porosity, fertility and productivity of the soil systems. The microbial imaging analysis was performed after 7 years from soil modification. It was shown that the anthropogenic soil hosted a mature microbial community useful for land cultivation. Further studies will be performed to test their potential to improve the aggregation of soils in Mediterranean environment.
- Thanks for your comments. All your suggestions were considered and the changes in the text were highlighted in yellow.
Essentially, there are no questions to the article. You may wish to pay attention to the following points:
- Section 3.4 presents data of statistical processing of data on physicochemical parameters and relative abundance of microbiota. In particular, Figure 5 shows a dendrogram obtained as a result of a hierarchical cluster analysis of soil samples, and Figure 6 shows, if not trends, then some similarity of the grouping of figurative points of soil samples on the principal component diagram. These data are not discussed. For example, Mant 3 is different from Mant 1 and Mant 2, while being closer to Fiume Palma Stream (I could be wrong) and probably lower. It is also possible that the authors believe that the observed differences are not significant or these nuances go away from the topic of the article.
- The spatial position of the three pedons could not be a factor driving any variance in the microbial profile, since the only factor dominant and regulating all the soil variables is the man action in generating such soils by transformation of the original vertisol using the described pedotechnique. The effects of any natural soil genetic factors have been canceled in the moment of the creation of the anthropogenic soil. This explanation has been added to the discussion section (L997-1002). Thanks anyway for this detailed comment.
- The use of "marly limestone", in my opinion, is not correct. Based on the composition, marl is a mixture of clay and carbonate (most often calcite). That is, simplistically, marly limestone = mixture of (limestone + mudstone) and limestone, and if you open the brackets, then it is either calcareous marlstone or argillaceous limestone. Perhaps from the point of view of structure or facies, or traditional names of fertilizers, the use of "marly limestone" will seem convenient to you, but at the end of this path, the terminology will cease to be universal.
- The reviewer is right. Terms have been recently revised. So, the correct term is ‘calcareous marls’ (L149,751,759,767).
Specific remarks:
Lines 14-15. If numerical values of calcium carbonate content in soil are given for a control soil, the corresponding numerical value for anthropogenic soil should probably be given.
- Missing information added (L16).
Line 33. "Anthrosol".
- Changed (L34).
Line 36. focused.
- The beginning of the sentence was modified by the English reviewer during language check (L37).
Lines 84-85. "The lithology dates back to the Pliocene and Miocene consisting of <sediments type1>, <sediments type2> and sediments." There is an error in this sentence. If it makes no sense to describe the lithology in more detail, then at the end of the sentence "and sediments" can be replaced by "and other sediments".
- Corrected into ‘gypsum, marls, limestones and alluvial deposits’ (L98-99).
Line 85. marly gypsum - Perhaps you mean gypsum marl?
- They are both, i.e. …gypsum, marls…(L98-99).
Line 98. You might want to change the phrase so that the word "depth" doesn't appear twice.
- “deep ploughing” was replaced by “ploughing” (L151).
Line 107. If there is a need to emphasize the collection of each pedon in duplicate, then a brief explanation should probably be given. For example, one sample was selected for physical and chemical studies, and the second sample was selected for microbiological studies. If it could be assumed that two specimens were selected to increase the volume of the pedon sample, then the question of the representativeness of the samples, more precisely, the volume of samples and the homogeneity of the substance in terms of the studied parameters, would be relevant.
- A given soil sample (bulk soil) was used for both physicochemical and microbiological analyses. As reported in M&M (L216), 20 g were used for microbiological determination from 200 g collected from the same bulk soil used for the other analyses. The only differences is that soil for microbiological investigations was aseptically collected (L201). We are opened to further suggestions regarding this point, If the reviewer thinks that these information are not enough for readers.
Lines 107-108. If there was a reason why soil samples affected and not affected by pedotechnics were taken specifically at these same depths - 0-10, 10-30, 30-50 cm, then perhaps it would be appropriate to devote a couple of lines to its description here (or provide a link to the description of the method). Perhaps it should be clarified that the A B C horizons do not reflect the structure of the soil profile?
- A, B, and C are depths, not horizons, we specify now that question and also the fact that they are part of the same ploughed horizons in the soil profiles. Depths were fixed to have a certain homogeneity in the sampling that could affect the distribution of microorganisms along with the soil profile depth. However, the capital letters A, B and C were replaced by the Roman numbers I, II and III, respectively, throughout the text and in all tables and figures.
Line 116. a portable conductivity-meter cond 7 - May be Cond 7 or COND 7? If you do not want to write the name of the device with a capital letter, then perhaps you would prefer to write it in quotation marks?
- Modified as “Cond 7” (L195).
Line 269. Space before Results.
- Space added (L501).
Line 388. pH.
- “pHs” was replaced by “values of pH” (L756).
Line 526. Typo - extra number 5.
- Removed (L1082).

Round 2
Reviewer 2 Report
All issues were adequately addressed, the alpha-diversity analyses, PCA and correlation analyses were adequately performed and analyzed.